# HIP (HPA-screening in pregnancy) study: protocol of a nationwide, prospective and observational study to assess incidence and natural history of fetal/neonatal alloimmune thrombocytopenia and identifying pregnancies at risk

Dian Winkelhorst [1,2] Thijs W de Vos [2,3] Marije M Kamphuis,[4] Leendert Porcelijn,[5] Enrico Lopriore,[3] Dick Oepkes,[1] C Ellen van der Schoot,[2,6] Masja de Haas [7,8]

CEvdS and MdH contributed equally.

For numbered affiliations see end of article.

**Correspondence to**
Dr C Ellen van der Schoot;
e.vanderschoot@sanquin.nl

## ABSTRACT

**Introduction** Fetal and neonatal alloimmune thrombocytopenia (FNAIT) may lead to severe fetal or neonatal bleeding and/or perinatal death. Maternal alloantibodies, targeted against fetal human platelet antigens (HPAs), can result thrombocytopenia and bleeding complications. In pregnancies with known immunisation, fetal bleeding can be prevented by weekly maternal intravenous immunoglobulin infusions. Without population-based screening, immunisation is only detected after birth of an affected infant. Affected cases that might have been prevented, when timely identified through population-based screening. Implementation is hampered by the lack of knowledge on incidence, natural history and identification of pregnancies at high risk of bleeding. We designed a study aimed to obtain this missing knowledge.

**Methods and analysis** The HIP (HPA-screening in pregnancy) study is a nationwide, prospective and observational cohort study aimed to assess incidence and natural history of FNAIT as well as identifying pregnancies at high risk for developing bleeding complications. For logistic reasons, we invite rhesus D-negative or rhesus c-negative pregnant women, who take part in the Dutch population-based prenatal screening programme for erythrocyte immunisation, to participate in our study. Serological HPA-1a typing is performed and a luminex-based multiplex assay will be performed for the detection of anti-HPA-1a antibodies. Results will not be communicated to patients or caregivers. Clinical data of HPA-1a negative women and an HPA-1a positive control group will be collected after birth. Samples of HPA-1a immunised pregnancies with and without signs of bleeding will be compared with identify parameters for identification of pregnancies at high risk for bleeding complications.

**Ethics and dissemination** Ethical approval for this study has been obtained from the Medical Ethical Committee Leiden-The Hague-Delft (P16.002). Study enrolment began in March 2017. All pregnant women have to give informed consent for testing according to the protocol. Results of the study will be disseminated through congresses and publication in relevant peer-reviewed journals.

**Trial registration number** NCT04067375.

## Strengths and limitations of this study

► The human platelet antigen-screening in pregnancy study is a unique prospective and completely non-interventional screening study with a large cohort that enables assessing the true natural history of fetal and neonatal alloimmune thrombocytopenia (FNAIT).

► The unique infrastructure in the Netherlands with one national referral laboratory for FNAIT (Sanquin, Amsterdam) collaborating with the national fetal therapy centre (Leiden University Medical Centre, Leiden) will result in complete data and focus on both laboratory and clinical parameters.

► A limitation of the study is that we rely on the clinical judgement of bleeding tendency after birth and do not obtain cord blood platelet counts or perform routine neonatal cerebral ultrasounds. Therefore, we may still underestimate disease prevalence due to subclinical cases.

## INTRODUCTION

Fetal and neonatal alloimmune thrombocytopenia (FNAIT) is the most frequent cause of severe thrombocytopenia in term-born infants.[1 2] FNAIT is caused by the production of maternal alloantibodies against the paternally derived, fetal human platelet antigens (HPAs). Clinical consequences can vary from an asymptomatic thrombocytopenia to minor

skin haemorrhage, such as haematoma or petechiae, or ultimately severe internal organ and intracranial haemorrhage (ICH).[3 4] Bleeding complications that, in subsequent pregnancies, can be effectively prevented by weekly administration of intravenous immunoglobulins (IVIg) to the mother.[5] The vast majority of cases with (severe) clinical consequences are caused by maternal alloantibodies targeted against fetal HPA-1a.[6–8] FNAIT is considered to be the platelet counterpart of haemolytic disease of the fetus and the newborn (HDFN) because of their similar pathophysiologic fundaments. In this comparison, HPA-1a, that causes 90% of the ICH caused by FNAIT, is regarded to be the equivalent of rhesus D (RhD) of the red blood cell (RBC) in HDFN.[8] Important differences, however, exist as well. First, whereas RhD is only expressed on RBCs, the HPA-1a epitope expressed on platelets is also present on the membrane of endothelial cells and syncytiotrophoblast cells.[9 10] Second, whereas RhD is mainly a problem of second or subsequent incompatible pregnancies, more than half of the severe cases of HPA-1a-mediated FNAIT already occur in firstborn children.[4 11] For decades, the possibility of prevention of FNAIT by population-based screening for HPA-1a is discussed, in analogy to the RhD prophylaxis and erythrocyte immunisation screening.[12–14]

Careful evaluation of the feasibility, benefits, harms and cost-effectiveness of a possible FNAIT screening programme showed that knowledge is missing on different aspects of the disease. First, despite a couple of large prospective cohort studies, no data exist on the natural history of the disease. Most of the large prospective, screening studies performed, were not only observational, but included some kind of intervention, thereby making it impossible to draw any firm conclusion on the natural history of FNAIT.[15–19] Further, more accurate estimates of incidence and prevalence of the disease in the Dutch population need to be known. One of the most important differences, making it hard to implement a programme similar to the antenatal screening programme for erythrocyte immunisation, is the lack of tools to identify pregnancies at high risk for developing bleeding complications. Detecting HPA-1a negative women and further HPA-1a alloimmunised pregnancies can be done easily. When alloimmunisation is detected in HDFN, several parameters, laboratory as well as clinical, are available to assess disease severity and to predict which cases would benefit from treatment. For example, RBC alloantibody titre and functional assays such as an antibody-dependent cellular cytotoxicity assay can be performed, followed in preselected cases by estimation of fetal anaemia by Doppler-based assessment of flow velocity in the middle cerebral artery of the fetus. In this way, high-risk cases are identified that most likely benefit from fetal blood sampling (FBS), followed by an intrauterine transfusion.[20] Treating all HPA-alloimmunised pregnancies with IVIg would lead to a considerable and undesirable overtreatment. So, identification of HPA-alloimmunised pregnancies at high risk for disease, like in HDFN, would be preferable as well. FBS to determine fetal platelet count and if necessary administer intrauterine platelet transfusion can be performed in these pregnancies as well. However, in potentially thrombocytopenic fetuses, this is a risky procedure with a high rate of associated complications.[5] Unfortunately, no noninvasive laboratory or clinical diagnostic tests to select HPA-alloimmunised pregnancies that would benefit from treatment are applicable in a clinical setting.

To obtain information necessary to judge the effectiveness and feasibility of a potential population-based screening, we designed the HPA-screening in pregnancy (HIP) study. With the HIP study, we aim to collect data on the incidence of HPA-1a alloimmunisation and clinically relevant FNAIT in the Netherlands. The study will be completely observational. This way we will be able to conclude on the natural history of FNAIT. Ultimately, by comparing test characteristics of blood samples from pregnancies with and without clinical manifestations of bleeding, we aim to develop one or more diagnostic tools, allowing more effective and personalised management by selecting pregnancies at high risk for bleeding complications that have the highest chance to benefit from antenatal preventive treatment with IVIg. This would not only be desirable in current management of FNAIT but also especially in potential future screening setting.

## METHODS AND ANALYSIS
### Study objectives
The primary objective of this study is to determine incidence of HPA-1a alloimmunisation and the incidence of clinically relevant HPA-1a-induced FNAIT in the Netherlands. Clinically relevant FNAIT will be defined as minor bleeding (haematoma, bruising, petechiae or small visceral bleeding) and severe bleeding (ICH or internal organ haemorrhage) with the presence of an anti-HPA-1a alloantibody. Additionally, as secondary objective, we aim to collect a set of blood samples that can contribute to the development of a risk assessment model to be used as a diagnostic tool enabling the identification of alloimmunised pregnancies that are at high risk of developing bleeding complications.

### Study design
The HIP study is a nationwide prospective and observational cohort study, conducted in all settings of obstetric care in the Netherlands, for a period of 2.5 years.

### Patient and public involvement
In 2008, the Ministry of Health, Welfare and Sport (in Dutch: Ministerie van VWS) gave instructions to investigate preventive interventions for 27 significant health problems that could be cost-effective. As a result, the National Institute for Public Health and the Environment (in Dutch: RIVM) published a report stating that antenatal screening for FNAIT would be cost-saving, but they advised that more knowledge on natural history of the disease and treatment of detected cases should

be obtained to support possible implementation of screening.[21] Also, the RIVM was involved in the design of the study. There was no further involvement of patients or public in the recruitment or the conduct of the study.

## Study population

For logistic purposes, RhD-negative or rhesus c (Rhc)-negative pregnant women were selected for enrolment in the HIP study. As part of the Dutch prenatal screening programme for infectious disease and erythrocyte immunisation (in Dutch: PSIE), these women are offered a free of charge red cell antibody screening and/or fetal RhD typing at 27 weeks gestation. For this, 9 mL EDTA anticoagulated blood is drawn by their midwife or at certified, local laboratories all over the Netherlands (n=±90) and transported to the Sanquin laboratory in Amsterdam by regular surface mail or private courier service. The programme has a voluntary participation grade of 99%.[22 23] With approval of the RIVM, that organises this population screening programme, left-over material can be used for the HIP study for HPA-1a typing and stored for further antibody testing after informed consent.

### Inclusion criteria

Prior to enrolment, participants have to fulfil these following criteria:

► Pregnant women participating in the currently implemented prenatal screening programme for erythrocyte immunisation and who are typed RhD or Rhc negative.
► Ability to make an informed decision on participating in the population screening programme as well as in the HIP study.

### Exclusion criteria

► Cases with insufficient material to perform HPA-1a typing by ELISA.
► Cases with known HPA-1a alloimmunisation.

## Participating centres

All obstetric care centres, hospitals, midwifery practices as well as general practices that provide obstetric care, in the Netherlands, are able to enrol pregnant women to participate in the HIP study. In order to ensure that obstetric caregivers were equipped to inform and counsel pregnant women, communicatory symposia were organised at six locations all over the Netherlands. Additionally, an informational leaflet was produced in different languages (Dutch and English on paper; Spanish, Arabic, Turkish and Polish digitally available; online supplementary material 1). Two informational videos were made informing on FNAIT as well as the HIP study. Finally, a website was created containing news and information about the HIP study (www.HIPstudie.nl).

## Study outcomes

The main study parameters/primary endpoints are

► Incidence of HPA-1a negativity in the RhD-negative or Rhc-negative pregnant population in the Netherlands at 27 weeks of pregnancy.

Incidence of HPA-1a alloantibodies in the tested population at 27 weeks of pregnancy.
Incidence of clinically relevant HPA-1a-mediated FNAIT; classified as mild or severe FNAIT
– Severe FNAIT
  – ICH.
  – Internal organ haemorrhage.
– Mild FNAIT
  – Neonatal bleeding signs other than ICH or internal organ haemorrhage: haematoma, bruising, petechiae, purpura, mucosal or visceral bleeding.
  – Thrombocytopenia for which treatment was administered (platelet transfusion or IVIg) or for which clinical observation was performed.

Our secondary study parameters/endpoints are:

► Neonatal treatment for thrombocytopenia: platelet transfusion (with random-donor platelets vs compatible platelets), IVIg, RBC transfusion.
► Neonatal morbidity: small for gestational age, infection, hours/days in hospital (neonatal intensive care unit vs medium care), need for additional treatment, congenital abnormalities, other causes causing increased bleeding tendency.
► Neonatal laboratory findings: platelet count, haemoglobin, C-reactive protein.

## HIP study procedure

As part of the prenatal screening programme for erythrocyte immunisation, an EDTA tube of blood of RhD-negative and Rhc-negative pregnant women will be sent to Sanquin at 27 weeks gestation. These women are eligible for enrolment in the HIP study and will be informed about the study and asked for consent by their obstetric caregivers. This consent or decline of participation is added to the regular laboratory request form for the 27th week assessment, that is, already sent to Sanquin with each tube of blood. No additional blood will be drawn for the HIP study. Once the tubes of blood are sent to Sanquin, the consent is either received digital or on paper, depending on the route and location (various hospitals, midwifery practices and local laboratories).

The procedures that are performed after consent and enrolment in the HIP study can be divided into four separate phases, depending on the time in and after pregnancy (figure 1).

### Phase I

After regular screening, authorisation and correspondence of the results for the prenatal screening programme for erythrocyte immunisation, the tubes are made available for the HIP study. For the HIP study, the platelet-containing plasma of the stored blood tubes is serologically typed for HPA-1a, using a sandwich ELISA. In short, 20 µL of plasma containing platelets will automatically be pipetted into microtiter plates that have been coated with a monoclonal antibody CLBthromb/1 (C17) directed against glycoprotein IIIa, at a concentration of

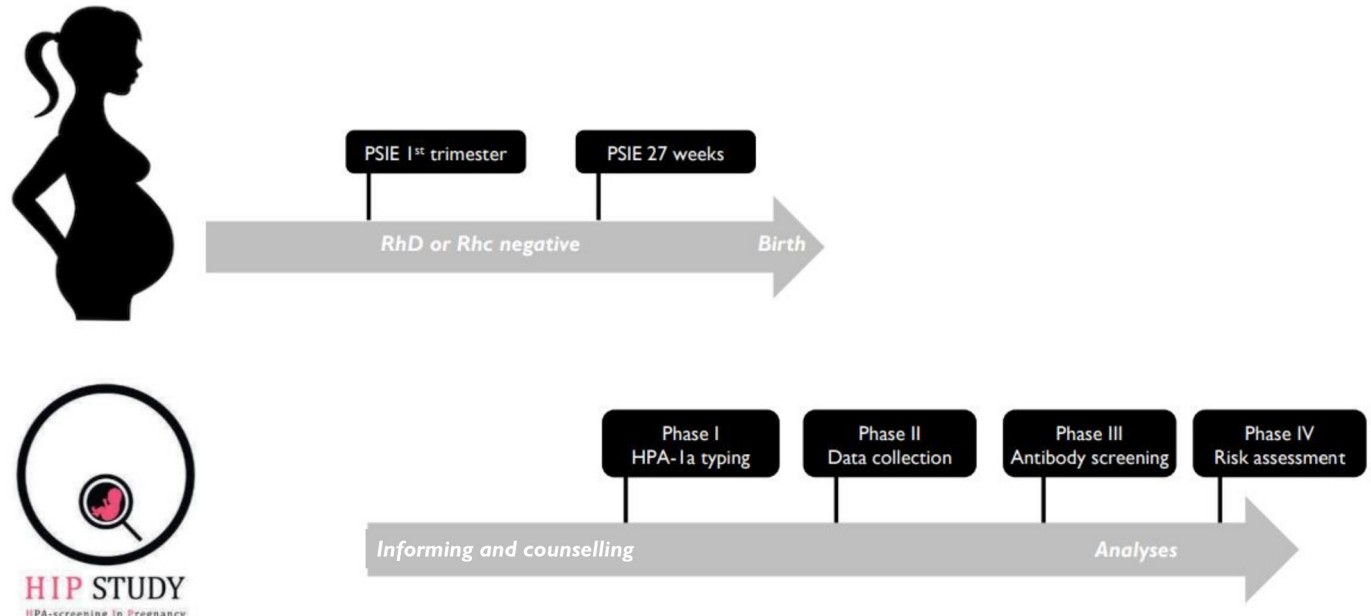

**Figure 1** Schedule of selection, enrolment and tests in the HIP study. HIP, HPA-screening in pregnancy; HPA, human platelet antigen; PSIE, prenatal screening of infectious diseases and erythrocyte immunisation; RhD, rhesus D; Rhc, rhesus c.

3 µg/mL to capture all platelets from the plasma. Then horseradish peroxidase-conjugated (HRP-conjugated) B2G1, an antibody targeting HPA-1a, will be added and plates will be centrifuged and incubated for 45 min. Finally, after washing of the plates, HRP-substrate solution will be added for 15 min, and after stopping of this reaction, the reactions will be quantified using an ELISA reader (Biochrom Anthos, Cambridge, UK). This HPA-1a ELISA was specifically designed for the HIP study, thus for quick and high-throughput screening. All samples with an ELISA value below a defined optic density are called HPA-1a negative. The HPA-1a typing result is supported with an allelic discrimination PCR assay. Plasma and buffy coat of samples that are typed HPA-1a negative will be stored at −20°C, using only a study number. Additionally, for each HPA-1a negative case, material of one HPA-1a positive control will be stored simultaneously.

Because this first phase comprises serological HPA-1a typing, which is performed with fresh material, and a delay in the arrival of consent forms might exist, this phase is performed with all samples from pregnant women who did not decline participation for the HIP

study. All consecutive phases, such as antibody screening, risk-assessment development and clinical data retrieval, are solely performed in case of informed consent for the HIP study.

### Phase II

Of all samples stored with consent, obstetric caregivers will be contacted to obtain clinical information. An overview of these clinical parameters is provided in figure 2. The clinical data will be stored in a secured digital database, designed by the Leiden University Medical Centre (LUMC), called ProMISe. First, study numbers of HPA-1a negative cases and HPA-1a positive controls with corresponding obstetric caregivers are entered into the database. Then, for each case, ProMISe randomly generates a code. Thereafter, obstetric caregivers will receive a secured digital invitation to add clinical data to a digital case report form (CRF) for the cases from their practice. This secured invitation contains the initial personal data for the sample sent for the erythrocyte immunisation screening programme together with the code generated by ProMISe (Project Manager Internet Server). This

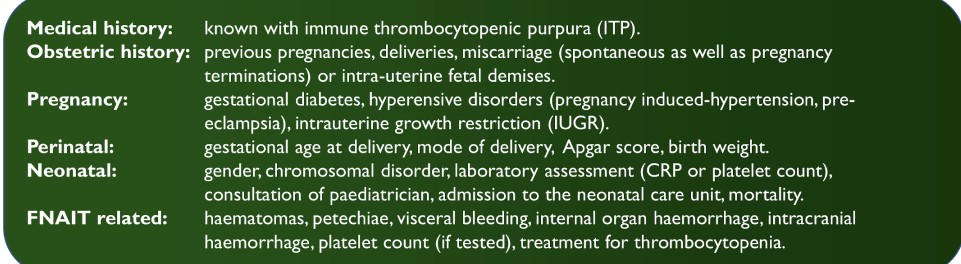

**Figure 2** Clinical parameters. These clinical parameters will be collected in the human platelet antigen-screening in pregnancy study. CRP, C-reactive protein.

unique code can be entered online by the caregiver to open the digital CRF and fill in the clinical data. Clinical data are stored in ProMISe, only by anonymous study codes. This way, no personal information is being transferred or entered in our database, nor is the obstetric caregiver in possession of a key that links the anonymous study number to personal information, nor does the caregiver know whether their patients or clients are HPA-1a negative or positive.

## Phase III

The next step is to evaluate the incidence of alloimmunisation. Of all HPA-1a negative women who gave consent for the HIP study, we will use the stored left-over plasma to screen for HPA-1a alloantibodies. For antibody screening, the Pak Lx assay, a qualitative immunoassay, will be used, according to the manufacturers' recommendations (LIFECODES Pak Lx Assay, Immucor GTI Diagnostics, Norcross, USA). In short, plasma samples are incubated with reconstituted beads and for the removal of unbound antibodies, the beads are washed. Next a conjugate (anti-human IgG antibody conjugated to phycoerythrin) is added and incubated with the sample for 30 min at room temperature. Finally, the Luminex 200 instrument is used to analyse the data. The advantage of this assay is that it is quick and uses only a small amount of plasma, so there will be enough left-over for further testing in phase IV.

## Phase IV

Combining the results from phase II to phase III will enable us to select cases of alloimmunisation with and without clinical manifestations of FNAIT to identify possible parameters to predict the development of (severe) bleeding complications. For this, we will be testing different laboratory parameters as well as clinical parameters (figure 2). Laboratory parameters that will be tested to assess risk at bleeding complications are HLA-DRB3*0101 status, antibody level, Fc-core glycosylation and FcγRIII-binding index, endothelial cell binding, endothelial cell function.[24–28]

## Sample size calculation

The HIP study was designed to assess the incidence of clinical relevant FNAIT in pregnant women in the Netherlands. Therefore, the incidence of ICH in HPA-1a immunised cases was compared with HPA-1a positive women. The estimated risk of ICH in immunised cases was 3%. For our power calculation, we took a marge of 1% on the estimated incidence of ICH in FNAIT.[19 29] In our control group, we assumed a risk on symptomatic ICH of 4.9 in 10.000 (0.05%).[30] To achieve a power of 80% at an alpha level of 5%, we calculated that a total study population of 2400 pregnant women is needed. Within this calculation, we took into account the unequal distribution between HPA-1a positive controls and immunised cases. We considered 5% of our total study population consists of immunised cases, which means that we needed to include 120 immunised cases. Calculations were performed using logistic regression model making use of PASS sample size software 11.

Each year, approximately 60 000 RhD-negative or Rhc-negative pregnant women participate in the prenatal screening programme for erythrocyte immunisation and are therefore eligible for enrolment in the HIP study. To include 120 immunised cases, we need to include 1200 HPA-1a negative women (immunisation rate of approximately 10%).[29] Because 2.1% of the Caucasian population is HPA-1a negative, the total study population should exist of 60 000 pregnant women (table 1). Based on the previous experience with earlier nationwide research on the detection and prevention of pregnancy immunisation (Opsporing en Preventie Zwangerschaps Immunisatie, OPZI studie in Dutch) and the highly positive attitude towards potential HIP women expressed in our previous study, the expected enrolment was 50%.[31 32] This would correspond with a study period of 2 years.

## Statistical analysis

Clinical data will be entered into a validated data capture system, provided and designed by the LUMC. The system is protected by password and contains internal quality checks to identify inaccurate or incomplete data. Laboratory data will be entered in a separate password-protected database by independent technicians, inaccessible to the researchers. Both clinical and laboratory data will be combined and further data management and analysis will be performed using SPSS V.23.0 and Graphpad V.8.0. An interim analysis after 1 year will be performed.

| Table 1 | Estimated cases in HIP study | | | |
|---|---|---|---|---|
| | % | Incidence | Cases in the Netherlands Total pregnancies n=1 70 000 | Cases during study period Total included n=60 000* |
| HPA-1a negative | 2.1 | 1:50 | 3570 | 1260 |
| HPA-1a antibodies | 10 | 1:400 | 428 | 126 |
| Severe FNAIT | 30 | 1:1300 | 129 | 36 |
| ICH | 10–30 | 1:12 500 | 13 | 3–4 |

*Assuming 50% enrolment of the 60 000 RhD/Rhc negative women each year, for 2 years.
FNAIT, fetal and neonatal alloimmune thrombocytopenia; HIP, HPA-screening in pregnancy; HPA, human platelet antigen; ICH, intracranial haemorrhage.

### Ethics and dissemination

The introduction of an antenatal screening programme requires a careful balance between benefit and potential harm. To investigate the true natural history of FNAIT, we aimed to collect data from pregnancies without additional interventions based on screening test results. This observational non-intervention design is ethically challenging. It would be unethical to share the antibody screening results with pregnant women but withhold them from therapy, therefore antibody screening will be performed far after due date. There will be no direct beneficial effect for pregnant women participating in the HIP study, pregnant women will be informed by their caregivers about this before they give consent to our study. Ethical approval for this study has been obtained from the Medical Ethical Committee Leiden-The Hague-Delft (P16.002).

Patient recruitment started in March 2017 and the study is planned to close to recruitment on the spring/summer of 2019. However, to ensure the inclusion of 1000–1500 HPA-1a negative women, the inclusion period might take longer. Accurate predictions on the duration of the study will be made after interim analysis at 1 year. Results will be published in relevant scientific journals and be disseminated in international conferences when inclusion and clinical data collection is finished.

### DISCUSSION

FNAIT can cause severe bleeding complications in fetuses and neonates, with a high risk of associated morbidity and mortality.[33] A preventive antenatal treatment, that effectively prevents these bleeding complications from occurring, is available.[5] In current practice, this prevention is only available in pregnancies with known alloimmunisation, usually after a previously affected child. To prevent these first cases as well, timely detection by prenatal and population-based screening is necessary.

Current lack of prospective non-interventional studies providing data on natural history of the disease as well as a reliable risk assessment tool to identify alloimmunised pregnancies that are at high risk for developing bleeding, complicates the implementation of such population-based screening. The aim of the HIP study is to gather this missing knowledge necessary to adequately evaluate the potential efficacy and feasibility of prenatal population-based screening in order to timely detect and prevent FNAIT-related complications. With the current study design and logistics, making use of the current national screening programme for RBC immunisation with a participation grade of 99.1%. We do not think that selection of RhD-negative and Rhc-negative women would influence the outcome of our study (ie, immunisation rate or bleeding symptoms). RhD and Rhc status has never been associated with platelet immunisation during pregnancy and inheritance is unrelated since the RhD and Rhc genes are located on chromosome 1 and the HPA-1 allele on chromosome 17. Therefore, we expect our results to give an adequate representation of the Dutch population of pregnant women.

A potential limitation of this study protocol is the lack of routine determination of neonatal platelet counts. However, the goals of potential screening and prevention of FNAIT are not to prevent a low platelet count as reflected as a laboratory result, but to prevent symptomatic disease, mainly ICH, with associated morbidity caused by FNAIT. However, routine neonatal cerebral ultrasound is not performed either. Therefore, cases of subclinical ICH without symptoms (such as convulsions or reduced consciousness) or additional bleeding manifestations might be missed, although in theory these might lead to developmental problems later in life. However, major ICHs detected in prospective studies that did perform routine cerebral ultrasound were cases that were symptomatic as well.[19 34]

Further underestimation might occur due to the fact that we will perform only a single screening for anti-HPA-1a alloantibodies, that is, at 27 weeks gestation. Immunisations that occur later in pregnancy or after delivery will not be detected. Also, immunisations that will result in complications and termination of pregnancy or IUFD before 27 weeks gestation will not be identified. However, in terms of assessing feasibility and cost-effectiveness of population-based screening, a slight underestimation is unquestionably preferred to an overestimation. On the contrary, we use another antibody screening method compared with earlier screening studies, which is possibly more sensitive compared with the monoclonal antibody immobilisation of platelet antigens (MAIPA) technique.[35] The PAK Lx assay was tested on a series with 100 cases with suspected FNAIT by our research group in 2014. In 26 of these cases, anti-HPA-1a was detected by MAIPA, all cases were detected and one more by PAKLx. Overall, to our knowledge, the HIP study will be a unique study to prospectively and observationally collect data on incidence and natural history of FNAIT by including this large number of pregnant women without performing any kind of intervention. Additionally, it will be the first study to be able to identify a unique and unbiased study group, that is, immunised pregnant women without disease and without intervention. This is the pre-eminent group to be used for the development of a risk-assessment platform in order to select immunised pregnancies that are at high risk to develop bleeding complications and would therefore benefit from antenatal preventive measures, such as IVIg treatment.

**Author affiliations**
[1]Obstetrics, Leiden University Medical Center, Leiden, Zuid-Holland, The Netherlands
[2]Department of Experimental Immunohematology, Sanquin, Amsterdam, The Netherlands
[3]Pediatrics, Leiden University Medical Center, Leiden, Zuid-Holland, The Netherlands
[4]Obstetrics and Gynaecology, Onze Lieve Vrouwe Gasthuis, Amsterdam, Noord-Holland, The Netherlands
[5]Immunohaematology Diagnostics, Sanquin Blood Supply Foundation, Amsterdam, Noord-Holland, The Netherlands

[6]Landsteiner Laboratory, Academic Medical Center Amsterdam and Department of Experimental Immunohematology, University of Amsterdam and Sanquin, Amsterdam, The Netherlands

[7]Department of Immunohaematology Diagnostics, Sanquin, Amsterdam, The Netherlands

[8]Immunohaematology and Blood Transfusion, Leiden University Medical Center, Leiden, Noord-Holland, The Netherlands

**Contributors**   DW, DO, CEvdS and MdH designed the study; MMK, LP and EL commented on the design of the HIP study; DW, TdV, LP, DO, CEvdS and MdH wrote the study materials and coordinated the study design, wrote and reviewed this paper.

**Funding**   This project was funded by Landsteiner Foundation for Blood Transfusion Research (1440).

**Competing interests**   None declared.

**Patient and public involvement**   Patients and/or the public were involved in the design, or conduct, or reporting, or dissemination plans of this research. We refer to the Methods section for further details.

**Patient consent for publication**   Not required.

**Provenance and peer review**   Not commissioned; externally peer reviewed.

**ORCID iDs**
Dian Winkelhorst http://orcid.org/0000-0002-6538-5752
Thijs W de Vos http://orcid.org/0000-0002-3653-3234
Masja de Haas http://orcid.org/0000-0002-7044-0525

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
