## [Reviewer comments · BMJ Open]

ARTICLE DETAILS

TITLE (PROVISIONAL)	HIP-study (HPA-screening In Pregnancy): Protocol of a nationwide, prospective and observational study to assess incidence and natural history of fetal/neonatal alloimmune thrombocytopenia and identifying pregnancies at risk
AUTHORS	Winkelhorst, Dian; de Vos, Thijs; Kamphuis, Marije M.; Porcelijn, Leendert; Lopriore, Enrico; Oepkes, Dick; van der Schoot, C. Ellen; de Haas, Masja

VERSION 1 – REVIEW

REVIEWER	Marzena Debska Centre of Medical Postgraduate Education ul. Marymoncka 99/103 01-813 Warsaw 2nd Department of Obstetrics and Gynaecology
REVIEW RETURNED	06-Oct-2019

GENERAL COMMENTS	This is a very good study, which will certainly help to spread knowledge about FNAIT. I have a question, if the antibody level will be measured? What if the level will be high? Whether hiding this information will be ethical? Maybe it would be better to pass the information at least to the neonatologists so they could prepare themselves for the delivery of an thrombocytopaenic baby? What are you going to do if you will detect antibodies in a patient with ICH in the previous child?
---

REVIEWER	Heidi Tiller University Hospital North Norway, Norway and UiT The Arctic University of Norway, Norway
REVIEW RETURNED	10-Oct-2019

GENERAL COMMENTS	This manuscript describes the protocol of an ongoing screening study for HPA-1a alloimmunization in the Netherlands. The major strength of this study is the total lack of intervention, whereas the obvious limitation is the lack of neonatal platelet counts and cerebral ultrasound/CT/MRI in all newborns, and outcome data is only based on symptoms in the newborn. In general the study is well described and important limitations acknowledged by the authors. I miss a discussion regarding the use of RhDnegative/c negative pregnant women as being representative of all pregnant women. Are there any reasons to think they may not be representative? Further, possible limitations from using Pax Lx assay as a screening method to detect anti-HPA-1a antibodies instead of the gold-standard MAIPA methods should be
---

	addressed. In the abstract you state that: ..."In pregnancies with known immunisation, fetal bleeding is effectively prevented using maternal intravenous immunoglobulin...". The efficacy of antenatal IVIg to prevent ICH is not well documented, please rephrase to consider the lack of scientific evidence. Further, you state in the abstract that... "...Implementation is hampered by lacking knowledge on incidence...". This may be true for the Netherlands, but in general the incidence of HPA-1a alloimmunization/ FNAIT in a Caucasian population is well known and documented. Please rephrase. Minor comments: There are some grammatical and typing errors in the abstract. I also suggest to include secondary outcome analysis on birth weight and placenta weights, as these data should be available in this study, and would be very interesting to look at. Also, information on what laboratory method you will use to determine antibody levels is not given in the protocol.
--	--

REVIEWER	Ewa Brojer Institute of Hematology and Transfusion Medicine, Warsaw, Poland
REVIEW RETURNED	14-Oct-2019

GENERAL COMMENTS	Dear Dr Winkelhorst , After reading the protocol submitted to BMJ Open I would like to say that you initiated a very important study concerning FNAIT. I do however have some concerns about the description of the aim. In my opinion " to analyze complete natural history of FNAIT as well as the incidence of HPA-1a alloimmunisation and clinically relevant FNAIT" is a phrase that requires change as does the description of the project and the title - if the protocol is to be published in BMJ Open. I also think that it would be important to present the summary of any interim-analysis in the results section. According to the presented protocol the project started in March 2017 and the recruitment was planned to end in the spring/summer 2019. The interim-analysis was planned after 1 year. If the protocol was submitted for publication in September 2019 – it means that the inclusion period took longer. This should be referred to in the discussion section. Explanation: you plan to evaluate anti-HPA-1a only once at 27th weeks' gestation. You will not get the full picture of the natural history of FNAIT – since the mothers who develop anti-HPA-1a antibodies after 28th gw (both with and with no clinical symptoms of FNAIT) will be missed. In the study of Williamson et al HPA-1a antibodies were detected in 46 women (12%): first detected at or before 20 weeks in 27 pregnancies, between 21 and 32 weeks in eight, between 33 weeks and term in three, and postnatally in seven. It means that at least 3/46 =7% of women developed antibodies during pregnancy, but after 27thgw. In our large prospective PREVFNAT study of 24 246 non-selected pregnant women where they were tested for antibodies at 17-20, 28 th, 32nd and 38-40 gw in ~30% of mothers alloimmunised during pregnancy antibodies were detected for the first time at 28th or even at 38-40 gw. Some of them delivered babies with mild thrombocytopenia with no clinical signs of bleeding. In your study you will not identify alloimmunisation in the late period of gestation and alloimmunisation after delivery.
---

I must stress that your project is very important since it evaluates the realistic protocol for extending the routine examination of pregnant women for risk of hemolytic disease of the newborn into the identification of pregnancies at risk of fetal neonatal alloimmune thrombocytopenia (FNAIT). The value of the presented protocol is therefore very high and this should find reflection in the paper. Your study shows how many anti-HPA positive women will be identified by testing them once in 27th gw. This would answer the question of how many women should be treated to prevent bleeding. Bleeding frequency will also be analyzed both in HPA-1a negative - immunized and not immunized population and compared with the frequency of bleeding in the control group. Further analysis described in study phase IV could provide laboratory parameters to identify pregnancies at risk of bleeding.

I am not sure that such study can be called “prospective” – the antibody examination is performed retrospectively and not communicated to the patient.

I suggest the following major changes:

- Change of the protocol title: from “Protocol of a nationwide, prospective and observational study to assess incidence and natural history of fetal/neonatal alloimmune thrombocytopenia and identifying pregnancies at risk” to “ Protocol of a nationwide, observational study to assess the benefits of extending the prenatal screening program for erythrocyte immunization by HPA-1a typing and anti-HPA-1a diagnostics at 27th gw for identification pregnancies at risk of fetal neonatal alloimmune thrombocytopenia.
- Change of primary study objective from “... to assess incidence and natural history of FNAIT” to “to assess the effectivity of HPA-1a screening and anti-HPA-1a determination at 27 gw for identification pregnant women at high risk for developing bleeding complication”.
- Change of study outcome (page 8) to “incidence of HPA-1a antibody detection at 27th gw in the tested population”

It is my suggestion not to underline that HIP study is the first prospective and completely nointerventiononal screening study (page 3). It is true that most of the large prospective, screening studies were not only observational, but included some kind of intervention related to ethical reasons. You combined HPA-1a screening with routine screening for HDN risk and the results were not communicated to patients or gynecologists. This is of no harm to participants but of no benefit for them. I think it would be interesting to know how women and the society accept a study of this kind. This issue should be discussed after presentation of recruitment results.

Sincerely yours,
Ewa Brojer

HIP-study (HPA-screening In Pregnancy): Protocol of a nationwide, prospective and observational study to assess incidence and natural history of fetal/neonatal alloimmune thrombocytopenia and identifying pregnancies at risk
Elaboration of “No”: answers

1. Is the research question or study objective clearly defined? The study objectives should be changed (see the letter to dr Winkelhorst)
2. Is the abstract accurate, balanced and complete? The abstract should be corrected after changing the description of the aim of the study (see the letter to dr Winkelhorst)

	3. Is the study design appropriate to answer the research question? - It is strange: The study design is appropriate – but it can answer different questions than asked by the authors 6. Are the outcomes clearly defined? the outcomes should be changed(see the letter to dr Winkelhorst) 9. Do the results address the research question or objective? - There is no results section in the paper. It is only the description of the protocol. I suggest to include the results (see the letter to dr Winkelhorst) 10. Are they presented clearly? There are no results section 11. Are the discussion and conclusions justified by the results. The first part of the discussion page 11 – row 37 – 54 should be deleted (it belongs rather to the introduction section) I suggest to include the results of the interim-analysis which was planned after 1 year and to discuss them. It would be really interesting how the recruiting proceeds - to what extent pregnant women and gynecologists are interesting in the development of new strategies for pregnancy monitoring. 15. Is the standard of written English acceptable for publication? Sorry, I am not a native speaker. For me the written English is excellent.
--	--

VERSION 1 – AUTHOR RESPONSE

Reviewer 1: Marzena Debska

Institution: Centre of Medical Postgraduate Education

We would like to thank Marzena Debska for reviewing our protocol and sharing her enthusiasm about our study. In her commentary she addresses important ethical considerations in the design of our study. The Dutch ministry of Health, Welfare and Sport is not yet convinced about cost effectiveness of an antenatal screening programme for FNAIT however they acknowledge that it could turn out that antenatal screening on FNAIT would be cost-effective. To decide on the introduction on nationwide screening knowledge on two subjects should be obtained; natural history of disease and parameters to identify pregnancies at risk for severe outcome. One of the parameters that will be tested will be the HPA-1a antibody titre (as was suggested by the reviewer). We aimed to investigate both subjects, therefore it was important that no interventions related to the HPA-1a status or possible antibodies were performed during pregnancy.

We agree with Marzena Debska that it is unethical to perform antibody screening during pregnancy and withhold the test results from pregnant women or caregivers. To overcome this issue we decided to save left-over material from pregnant women and perform antibody screening far after the delivery of their child. If test results would have been shared with caregivers or pregnant women interventions to prevent possible burden of the antibodies could influence the course of disease. The cases that were included in our study would not have been screened since nationwide screening is not (yet) performed. Our study was designed in collaboration with the Dutch National Institute for Public Health and the Environment (RIVM) and approved by an ethical committee. We agree with Marzena Debska that this is an important point and therefore we addressed our ethical considerations with an extra paragraph (see below) in “Ethics and dissemination”.

{The introduction of an antenatal screening program requires a careful balance between benefit and

potential harm. To investigate the true natural history of FNAIT we aimed to collect data from pregnancies without additional interventions based on screening test results, this observational non-intervention design is ethically challenging. It would be unethical to share the antibody screening results with pregnant women but withhold them from therapy, therefore antibody screening will be performed far after due date. There will be no direct beneficial effect for pregnant women participating in the HIP study, pregnant women will be informed by their caregivers about this before they give consent to our study.}

Finally Dr Debska asks about what will be done with cases that have a history of ICH in the previous child. If FNAIT is known and the mother receives antenatal treatment the pregnancy will be excluded from our study. If FNAIT is not known this case will be included in our study and described like other cases.

Reviewer 2: Heidi Tiller

Institution: University Hospital North Norway

We would like to thank Heidi Tiller for reading our article, her complements about our protocol and her questions about our study. We do not think that selection of RhD/Rhc negative women would influence the natural course of FNAIT, we have added comments about the selection of RhD/Rhc negative pregnant women as being representative of all pregnant women in our discussion.

{We do not think that selection of RhD and Rhc negative women would influence the outcome of our study (i.e. immunization rate or bleeding symptoms). RhD and Rhc status has never been associated with platelet immunization during pregnancy and inheritance is unrelated since the RhD and Rhc genes are located on chromosome 1 and the HPA-1 allele on chromosome 17. Therefore we expect our results to give an adequate representation of the Dutch population of pregnant women.}

Second, Heidi Tiller asks some reflection on the possible disadvantage of using PAK Lx assay as a screening method to detect anti-HPA-1a antibodies instead the MAIPA technique which is currently the golden standard to detect platelet directed antibodies. The PAK Lx assay was tested on a series with 100 cases with suspected FNAIT, which was published in an article by Porcelijn et al in 2014. In 26 of these cases anti-HPA-1a was detected by MAIPA, all cases were detected and one more by PAKLx. We believe that PAK Lx is possibly more sensitive for HPA-1a antibody detection than the MAIPA technique. We added this consideration in the discussion of our article.

{On the contrary we use another antibody screening method compared to earlier screening studies which is possibly more sensitive compared to the MAIPA technique (Monoclonal Antibody Immobilization of Platelet Antigens). The PAK Lx assay was tested on a series with 100 cases with suspected FNAIT by our research group in 2014. In 26 of these cases anti-HPA-1a was detected by MAIPA, all cases were detected and one more by PAKLx.}

We agree with Heidi Tiller that maternal IVIg treatment to prevent bleeding in fetusses exposed to HPA-1a antibodies is not proven in a placebo controlled randomised trial. However, the evidence on the effectiveness of IVIg treatment is international acknowledged and it would be unethical to conduct a randomised placebo-controlled trial today. Recently maternal IVIg treatment as a first line treatment was acknowledged and recommended in the most recent international guideline on clinical management of FNAIT (Lieberman et al. 2019). There is a consensus among clinicians that treating these women with IVIg is efficient. We made some minor revisions to the statement in the abstract of our article.

In pregnancies with known immunisation, fetal bleeding {can be} is effectively prevented by {weekly} using maternal intravenous immunoglobulin (IVIg) infusions.

We agree with Dr Tiller that data on HPA-1a immunisation is well known but prospective data on clinical symptoms that was not influenced by any intervention is lacking. Collecting data on clinical symptoms instead of platelet count alone is unique to our study. We therefore choose to not rephrase this part of our article.

We revised our abstract carefully and corrected the grammatical and typing errors, probably we have overlooked some errors when adjusting the length of our abstract just before the first submission. We apologize for this inconvenience.

We would like to thank the reviewer for her suggestion to take birthweight and placenta weight as secondary outcome parameters. Placenta weight is not measured as standard of care and therefore we did not take this as an outcome parameter but we included the incidence of children small for gestational age as secondary outcome.

Neonatal morbidity: {small for gestational age}, infection, hours/days in hospital (NICU versus Medium Care), need for additional treatment, congenital abnormalities, other causes causing increased bleeding tendency.

Finally, Heidi Tiller asks which technique will be used to determine the antibody titre. This is now specified in the protocol by adding the following reference in paragraph: Phase IV.

Metcalfe P, Allen D, Chapman J, et al. Interlaboratory variation in the detection of clinically significant alloantibodies against human platelet alloantigens. *Br J Haematol* 1997;97:204-7.

Reviewer 3: Ewa Brojer
Institution: Institute of Haematology and Transfusion Medicine

We are grateful for the extensive review and letter of Ewa Brojer and her critical appraisal and suggestions for revision of our article. We would like to address her comments point by point.

Concerns about the description of the aim

We agree with the reviewer that since we perform antibody screening only once during pregnancy probably would lead to an underestimation of the burden of disease. We thank Dr Brojer for sharing her thoughts with us about the timepoint of immunisation. We agree that in the perfect study design we agree that antibody screening should be performed more than once however since the design of our study was observational and non interventional character of our study this was not possible. We think that we addressed this limitation the fourth paragraph of the discussion section. Since we will detect the majority of the FNAIT cases and we believe that we acknowledge the limitations of our study in our discussion section we think it would not be necessary to change the title and aims of our article.

Including results

The reviewer suggests to include results of the interim analysis in this paper. The aim of this paper was to publish our research protocol without any results since we wanted to establish our protocol and definitions before the publication of our results. Results will be published after clinical data collection of the whole cohort has been performed. We expect to finish data collection at the beginning of 2021. We clarified this in the dissemination section of our article.

Prospective study design

We thank Dr Brojer about her critical approach on the study design and her considerations about whether this design is prospective or not. We think the HIP study is prospective because we take our

sample (cohort) at 27th week during pregnancy and follow this group over time (longitudinally) until delivery.

Change of protocol title

We thank Dr Brojer for her suggestion on the adjustments of our protocol title. We both agree that this article concerns a: "Protocol of a nationwide, prospective and observational study to assess FNAIT". We agree that this data can be possibly used in the discussion on an antenatal screening programme but we would like to propose to keep our own title to underline the descriptive character of our study.

Change of study objective

We thank Dr Brojer for her suggestion for an adjustment of the study objective. As described in the previous paragraph we would like to propose remain our study objective unchanged to underline the descriptive character of our study. Our study was not designed to assess the efficacy of HPA screening during pregnancy but to describe the incidence and natural history of HPA-1a mediated FNAIT.

Change of study outcome

We agree with Dr Brojer our study outcome should be clarified by adding the timepoint of antibody screening in the study outcomes.

- Incidence of HPA-1a negativity in the RhD or Rhc-negative pregnant population in the Netherlands {at 27 weeks in pregnancy}
- Incidence of HPA-1a alloantibodies in the tested population {at 27 weeks in pregnancy}

Advise about describing the study

Dr Brojer suggests to not underline that the HIP study is the first prospective and completely noninterventional screening study. We agree with her on this point and removed this term from this phrase.

Strengths and limitations of this study

- The HPA-screening In Pregnancy (HIP) study is the first {a unique} prospective and completely non-interventional screening study with a large cohort that enables assessing the true natural history of fetal and neonatal alloimmune thrombocytopenia (FNAIT).

Discussion (last paragraph):

Overall, to our knowledge, the HIP-study will be the first {a unique} study to prospectively and observationally collect data on incidence and natural history of FNAIT by including this large amount of pregnant women without performing any kind of intervention. Additionally, it will be the first study to be able to identify a unique {and unbiased} study group ...

Ethical concerns

We agree with Dr Debska about the ethically challenging study design. We agree that it is unethical to perform antibody screening during pregnancy and withhold the test results from pregnant women or caregivers. To overcome this issue we decided to save left-over material from pregnant women and perform antibody screening far after the delivery of their child. To stress this important point we have chosen to address our ethical considerations with an extra paragraph (see below) in "Ethics and dissemination".

{The introduction of an antenatal screening program requires a careful balance between benefit and potential harm. To investigate the true natural history of FNAIT we aimed to collect data from pregnancies without additional interventions based on screening test results, this observational non-intervention design is ethically challenging. It would be unethical to share the antibody screening

results with pregnant women but withhold them from therapy, therefore antibody screening will be performed far after due date. There will be no direct beneficial effect for pregnant women participating in the HIP study, pregnant women will be informed by their caregivers about this before they give consent to our study.}

VERSION 2 – REVIEW

REVIEWER	Heidi Tiller University Hospital of North Norway, Norway
REVIEW RETURNED	08-Mar-2020

GENERAL COMMENTS	The paper has improved and I have no major concerns. There is one small typo in the abstract m&m: "AN HPA-1a positive control group..." (not a HPA...). In line 53-54 in the introduction, I miss a reference or two on the risks concerning FBS in thrombocytopenic fetuses.
--

REVIEWER	Brojer Ewa retired previously: Institute of Hematology and Transfusion Medicine Warsaw, Poland
REVIEW RETURNED	13-Mar-2020

GENERAL COMMENTS	Dear Dr de Haas and co-authors, I accept your explanations in response to my review and thank you for taking into account my suggestions. Best wishes, Ewa Brojer
--

VERSION 2 – AUTHOR RESPONSE

We want to thank also the reviewers for their revisions and suggestions. We revised the manuscript carefully and corrected spelling and grammar errors that we could identify. In addition to this we added references concerning the risks of FBS in response to the revisions suggested by reviewer 2.